# Bacterial Toxins from *Staphylococcus aureus* and *Bordetella bronchiseptica* Predispose the Horse’s Respiratory Tract to Equine Herpesvirus Type 1 Infection

**DOI:** 10.3390/v14010149

**Published:** 2022-01-14

**Authors:** Eline Van Crombrugge, Emma Vanbeylen, Jolien Van Cleemput, Wim Van den Broeck, Kathlyn Laval, Hans Nauwynck

**Affiliations:** 1Department of Translational Physiology, Infectiology and Public Health, Faculty of Veterinary Medicine, Ghent University, 9820 Merelbeke, Belgium; esvcromb.VanCrombrugge@UGent.be (E.V.C.); emma.vanbeylen@ugent.be (E.V.); 2Department of Internal Medicine and Pediatrics, Faculty of Medicine and Health Sciences, Ghent University, 9000 Ghent, Belgium; jolien.vancleemput@ugent.be; 3Department of Morphology, Medical Imaging, Orthopedics and Nutrition, Faculty of Veterinary Medicine, Ghent University, 9820 Merelbeke, Belgium; wim.vandenbroeck@ugent.be

**Keywords:** mucosal barriers, cell junctions, equine respiratory mucosal explants, EHV-1, α-hemolysin toxin, adenylate cyclase toxin, *Staphylococcus aureus*, *Bordetella bronchiseptica*

## Abstract

Respiratory disease in horses is caused by a multifactorial complex of infectious agents and environmental factors. An important pathogen in horses is equine herpesvirus type 1 (EHV-1). During co-evolution with this ancient alphaherpesvirus, the horse’s respiratory tract has developed multiple antiviral barriers. However, these barriers can become compromised by environmental threats. Pollens and mycotoxins enhance mucosal susceptibility to EHV-1 by interrupting cell junctions, allowing the virus to reach its basolateral receptor. Whether bacterial toxins also play a role in this impairment has not been studied yet. Here, we evaluated the role of α-hemolysin (Hla) and adenylate cyclase (ACT), toxins derived from the facultative pathogenic bacterium *Staphylococcus aureus* (*S. aureus*) and the primary pathogen *Bordetella bronchiseptica* (*B. bronchiseptica*), respectively. Equine respiratory mucosal explants were cultured at an air–liquid interface and pretreated with these toxins, prior to EHV-1 inoculation. Morphological analysis of hematoxylin–eosin (HE)-stained sections of the explants revealed a decreased epithelial thickness upon treatment with both toxins. Additionally, the Hla toxin induced detachment of epithelial cells and a partial loss of cilia. These morphological changes were correlated with increased EHV-1 replication in the epithelium, as assessed by immunofluorescent stainings and confocal microscopy. In view of these results, we argue that the ACT and Hla toxins increase the susceptibility of the epithelium to EHV-1 by disrupting the epithelial barrier function. In conclusion, this study is the first to report that bacterial exotoxins increase the horse’s sensitivity to EHV-1 infection. Therefore, we propose that horses suffering from infection by *S. aureus* or *B. bronchiseptica* may be more susceptible to EHV-1 infection.

## 1. Introduction

The appearance of respiratory symptoms in horses is greatly influenced by many environmental factors, including respiratory hazards [1]. These factors can impair the integrity of the horse’s respiratory mucosa and thereby can drive the infection and invasion of pathogens. Specifically, several respirable threats have already been described to impair and predispose the epithelium to viral infection in horses. For instance, it was demonstrated that pollen proteases could selectively and irreversibly alter cell junctions of columnar equine respiratory epithelial cells (EREC) and facilitate the invasion of equine herpesvirus type 1 (EHV-1) [2]. Moreover, it was shown that pretreatment of ex vivo respiratory mucosal explants and EREC with deoxynivalenol, a mycotoxin mainly present in equine feeds, damages respiratory epithelial integrity and also predisposes these cells to EHV1 infection [3]. So far, exotoxins originating from bacteria have not been investigated regarding their predisposing influence on EHV-1 infection. In this respect, *Staphylococcus aureus* (*S. aureus*) and *Bordetella bronchiseptica* (*B. bronchiseptica*) are particularly interesting bacteria. *S. aureus* is part of the normal equine nasal microbiome [4]. In Belgium and the Netherlands, the prevalence of *S. aureus* in equine nasal mucosa is estimated to be approximately 10% [5]. This facultative pathogenic and opportunistic bacterium will colonize the upper respiratory tract upon immunosuppression [6]. *B. bronchiseptica* is a free-living bacterium. Soil constitutes an environmental niche where the bacterium can proliferate and persist, which makes it omnipresent in the horse’s environment [7]. As a primary pathogen, *B. bronchiseptica* causes respiratory infection in the absence of a preceding viral infection [8,9]. Both bacteria produce and secrete a vast array of exotoxins upon colonization. Two of which are the α-hemolysin toxin (Hla) (originating from *S. aureus*) and the adenylate cyclase toxin (ACT) (produced by *B. bronchiseptica*) [10,11]. Various studies have pointed out the detrimental effect of these exotoxins on epithelial integrity in continuous cell lines. More precisely, both exotoxins cause the disruption of cell junctions [12,13,14,15]. 

EHV-1 is one of the major infectious agents affecting horses worldwide. The virus is responsible for respiratory disorders, abortion, neonatal foal death, and equine herpes myeloencephalopathy [16]. The main portal of EHV-1 entry is the upper respiratory tract. Efficient infection of the respiratory epithelium is a prerequisite for EHV-1 invasion through the basement membrane [17]. Following infection of the respiratory epithelium, EHV-1 exploits the local immune response by infecting monocytic cells and T-lymphocytes that are diapedizing through the infected epithelium [18,19,20]. Hereby, the virus crosses the basement membrane without being recognized. Subsequently, a cell-associated viremia is rapidly established, which allows the virus to spread to target organs such as the central nervous system or the uterus in gestating horses. Secondary replication in these organs often causes severe reproductive and neurological disorders in horses [21,22]. During invasion of the respiratory mucosa, the virus can also infect sensory nerve endings innervating the infected epithelium. Viral particles can therefore travel along axons to the trigeminal ganglia, where the virus establishes latency [23]. During periods of stress, the virus reactivates from its latent state and viral particles can travel back to the initial site of infection, where progeny virus is excreted [24]. The cycles of EHV-1 latency/reactivation contribute to a constant source of infection for potential new hosts. 

Interestingly, Van Cleemput and colleagues demonstrated that the disruption of cell junctions results in enhanced EHV-1 binding to explant basolateral surfaces [25]. Consequently, the authors hypothesized that EHV-1 targets a receptor located at the basolateral surface of the respiratory epithelial cells, which is only exposed when the integrity of the epithelium is compromised. 

Currently, no successful therapies are available against EHV-1 infection, and the control of EHV-1 infection mainly relies on management. Based on our current knowledge on the mechanisms of EHV-1 pathogenesis, it is clear that efficiently restricting EHV-1 replication in the respiratory mucosa might be a key strategy to fight the infection. In this regard, the control of environmental factors at the main portal of EHV1 entry might represent a promising approach to control primary viral infection. In this study, we aim to demonstrate whether bacterial exotoxins from *B. bronchiseptica* and *S. aureus* can similarly enhance EHV-1 replication in the equine respiratory mucosa, as previously shown with other respirable hazards. Specifically, we will investigate the effect of the Hla and ACT toxin on the morphology of the epithelium and its susceptibility to EHV-1 infection by using an in-house-developed ex vivo explant model. A better understanding and subsequent management of the plethora of respirable hazards that influence the horse’s environment and facilitate EHV1 infection might lead to the design of new and more effective therapeutic option to fight this infection.

## 2. Materials and Methods

### 2.1. The Equine Respiratory Explant Model

The proximal tracheas of six different healthy horses were collected at the slaughterhouse. An approval from the Ethical Committee from Ghent University was obtained to collect the horse tracheas from the slaughterhouse. The tracheas were immediately submerged in 500 mL of transport medium consisting of phosphate-buffered saline (PBS) with calcium and magnesium and supplemented with 100 U/mL penicillin (ThermoFisher Scientific, Paisley, UK), 0.1 mg/mL streptomycin (ThermoFisher Scientific), 0.1 mg/mL gentamicin (ThermoFisher Scientific), 0.1 mg/mL kanamycin (Merck, Darmstadt, Germany), and 0.25 µg/mL amphotericin B (ThermoFisher Scientific). An approval from the Flemish Food Safety Authority was obtained for transportation of horse tracheas from the slaughterhouse towards the laboratory. In the laboratory, the tracheas were rinsed once with approximately 200 mL of transport medium and the respiratory mucosa was cautiously removed from the underlying cartilage using sterile tweezers and a surgical blade. The mucosa was carefully placed in a Petri dish containing transport medium, and ciliary beating was assessed using an Olympus IX50 light microscope. After confirming sufficient ciliary beating, square tissue pieces of approximately 25 mm^2^ were prepared. Immediately afterwards, one explant per trachea was embedded in methocel and quick-frozen to assess the viability of the mucosa. By means of a terminal deoxynucleotidyl transferase-mediated dUTP nick-end labelling (TUNEL) staining (Roche, Basel, Switzerland), it was confirmed that transportation of the tracheas to the laboratory did not negatively affect the viability of the mucosal explants. For cultivation, the explants were placed epithelium upwards onto fine-meshed gauzes within 6-well plates, containing serum-free medium (50% DMEM (ThermoFisher Scientific)/50% RPMI (ThermoFisher Scientific)) supplemented with 100 U/mL penicillin, 0.1 mg/mL streptomycin (ThermoFisher Scientific), 0.1 mg/mL gentamicin (ThermoFisher Scientific), and 0.25 µg/mL amphotericin B (ThermoFisher Scientific). The explants were cultivated in an air–liquid interface, as previously described by Vandekerckhove et al. [26], for 18 h at 37 °C with 5% CO_2._

### 2.2. Pretreatment with Bacterial Toxins (Hla and ACT)

After 18 h of cultivation, the explants were removed from their gauzes and placed in a 24-well plate. To remove the mucus layer, the explants were thoroughly washed with serum-free medium, by flushing them with a P1000 pipette. Successful removal of mucus was evaluated with a light microscope. Additionally, sufficient ciliary beating was confirmed. Next, treatment with the bacterial toxins was carried out using the agarose model as previously published by Vairo et al. [27]. The apical surface of the epithelium was submerged in 1 mL serum-free medium containing either 0.1 µg/mL α-hemolysin toxin from *S. aureus* (Merck) or 0.5 µg/mL adenylate cyclase toxin from *B. bronchiseptica* (Merck) for 24 h at 37 °C. These concentrations were based on previous studies that showed cell junction disruption by the Hla and ACT toxins in CaCo-2 and VA10 cell lines [12,15]. As a positive control for the disruption of cell junctions, the epithelium was exposed to 8 mM of the calcium chelator ethylene glycol tetra-acetic acid (EGTA) (Merck) in PBS for 1 h at 37 °C. Explants were submerged in 1 mL of serum-free medium or PBS (negative control). After treatment with the bacterial toxins, the explants were carefully washed three times in serum-free medium within the agarose model. Afterwards, the explants were either fixed in 3.6% formaldehyde for 24 h, in preparation of morphological analysis by staining the sections with hematoxylin–eosin (HE), or embedded in methocel and quick-frozen for viability studies, or the explants remained in the agarose for subsequent EHV-1 inoculation. 

### 2.3. Assessment of Mucosal Morphology and Viability

Immediately after the 24 h treatment with the bacterial toxins, the explants were washed and fixed in 3.6% formaldehyde for 24 h for morphological analysis. Paraffin embedding of the explants was carried out using an automated system (STP 420D, Micron, Praran, Merelbeke, Belgium). Consecutive sections of 8 μm thick were cut, deparaffinized in xylene, rehydrated in descending grades of alcohol, stained with HE, dehydrated in ascending grades of alcohol and xylene, and finally mounted with DPX (DPX mountant, BDH Laboratory Supplies, Poole, UK). To evaluate the effect of the bacterial toxins on epithelial integrity and morphology, several parameters were evaluated by light microscopy. The percentage of intercellular space between the epithelial cells and the thickness of the epithelial layer were measured using Image J software (Image J, U.S. National Institutes of Health, Bethesda, ML, USA). 

For the determination of the percentage of intercellular space, a region of interest (ROI, i.e., the epithelium) was drawn manually in the “ROI manager tool”. Next, the threshold value to distinguish blank spaces from epithelial cells was determined, and the percentage of blank spaces between the cells (i.e., the intercellular space) was calculated. The thickness of the epithelium was measured using the Image J line-tool function. Further, the overall appearance of the epithelium (detachment of cells and presence of cilia) was visually assessed using light microscopy. 

To evaluate the viability of the explants upon treatment with the bacterial toxins, the explants were embedded in methocel immediately after treatment, quick-frozen, and stored at −70 °C until further processing. Cryosections of 16 µm were cut and fixed with 4% paraformaldehyde for 20 min at room temperature. Next, the sections were washed in PBS for 30 min. Afterwards, a TUNEL staining was performed following the manufacturer’s guidelines. TUNEL-positive cells were counted in five randomly chosen ROIs of 100 cells in the epithelium as well as the lamina propria, using a fluorescence microscope (Leica DM RBE microscope, Leica Microsystems GmbH, Heidelberg, Germany). 

### 2.4. EHV-1 Inoculation

In this study, we used the Belgian EHV-1 isolate 97P70. The virus was isolated from the lungs of an aborted fetus in 1997 [28]. The virus was grown and passaged on rabbit kidney 13 cells and was used in this study at its 6th passage. The explants were inoculated within the agarose model by exposing them to 1 mL of serum-free medium containing 10^6.5^ TCID_50_/_mL_ of the 97P70 for 1 h at 37 °C. After 1 h incubation, the explants were washed three times with serum-free medium and removed from the agarose. The explants were transferred back to their gauzes and incubated in serum-free medium in an air–liquid interface for 24 h, until collection. 

### 2.5. Evaluation of EHV-1 Infection by Immunofluorescent Stainings and Confocal Microscopy

After 24 h of incubation, the EHV-1 infected explants were frozen in methocel and stored at −70 °C until further processing. With a cryostat at −20 °C, 50 consecutive sections of 16 µm were cut per explant. The cryosections were loaded onto 3-aminopropyltriethoxysilane-coated (Merck) glass slides and fixed using 4% paraformaldehyde for 15 min at 4 °C. Afterwards, the tissue sections were washed three times in PBS. Permeabilization of the tissue was achieved by adding 0.1% Triton X-100 diluted in PBS for 10 min at room temperature. After washing the sections three times in PBS, an immunofluorescent staining to detect EHV-1 late proteins was performed, as previously described [29]. A biotinylated polyclonal horse anti-EHV-1 antibody was used as a primary antibody (dilution 1:20). The horse polyclonal anti-EHV-1 antibody was obtained by hyperimmunization of a horse [30]. Afterwards, the sections were incubated with streptavidin-FITC^®^ (ThermoFisher Scientific) (dilution 1:200). Hoechst 33342 (ThermoFisher Scientific) (dilution 1:100) was used to counterstain the nuclei. Finally, the slides were mounted with glycerol mounting medium, containing DABCO, and analyzed with a fluorescent microscope. The total number of plaques was counted on 50 cryosections (i.e., 4 mm^2^ epithelium) and the plaque diameter was measured using the Image J line-tool. Additionally, the percentage of EHV-1 late protein expression in the epithelium was determined using Image J software. The epithelium was defined as a region of interest by manually selecting it in the “ROI manager tool”. Next, the threshold value to distinguish the FITC-positive signal (i.e., EHV-1 late proteins detected by immunofluorescence) from the background signal was determined. Afterwards, the percentage of the ROI (i.e., the epithelium) that was positive for EHV-1 late proteins was calculated.

### 2.6. Statistical Analysis

Data are represented as mean values + SD of triplicate independent experiments. The data were analyzed for statistical significance using a one-way analysis of variance (ANOVA). The Tukey test was used as a post hoc test for multiple comparisons. Differences in results were considered statistically significant when the *p*-values were <0.05. Data were statistically evaluated with Prism 9 for macOS, version 9.3.0 (345). 

## 3. Results

First, we confirmed that treatment with 0.1 µg/mL of the Hla toxin or with 0.5 µg/mL of the ACT toxin did not induce cell death of respiratory epithelial cells or cells within the lamina propria, by means of TUNEL stainings (details are given in the Appendix A). 

### 3.1. The Hla and ACT Toxins Cause Morphological Changes in the Respiratory Mucosa, Characterised by a Reduction in Epithelial Thickness

#### 3.1.1. Treatment with the α-Hemolysin Toxin Causes a Reduction in Epithelial Thickness in Equine Respiratory Mucosal Explants

As shown in Figure 1A, treatment of the respiratory mucosal explants with 0.1 µg/mL Hla toxin did not result in a significant disruption of the cell junctions between the epithelial cells. The percentage of intercellular space in the epithelium after Hla treatment was 3.8 ± 0.6%, compared to 2.4 ± 1.5% in the untreated control. In contrast, the percentage of intercellular space was approximately 5-fold higher (14 ± 3.2%) in the positive control (EGTA treatment) than in the untreated control. A normal thickness of the epithelium (in nontreated control explants) is 58 ± 3.6 µm. The epithelial thickness decreased almost 2-fold to 34 ± 2.7 µm in Hla-treated explants compared to untreated ones (*p*-value < 0.001). Exposure to EGTA had no notable effect on the thickness of the epithelium (61 ± 4.6 µm compared to 58 ± 3.6 µm without treatment). Further, Hla treatment resulted in the detachment of single cells at the apical surface of the epithelium. Additionally, a partial loss of cilia was observed in explants upon Hla treatment when compared to untreated controls.

#### 3.1.2. Treatment with the Adenylate Cyclase Toxin Causes a Reduction in Epithelial Thickness in Equine Respiratory Mucosal Explants

The influence of the ACT toxin on mucosal morphology is shown in Figure 1B. Similarly to the Hla toxin, treatment with the ACT toxin did not induce an increase in intercellular spaces in the epithelium (2 ± 1%, compared to 1.5 ± 0.6% in the serum-free medium-treated control), while EGTA treatment clearly resulted in a disruption of the CJ (21 ± 3.6% intercellular space in the epithelium). Again, treatment with the bacterial toxin ACT resulted in a significant decrease in epithelial thickness from 64 ± 3.6 µm in normal mucosa to 39.3 ± 2.0 µm in ACT-treated mucosa (*p*-value < 0.001). The thickness of the epithelium was not altered upon EGTA treatment (65.6 ± 5.9 µm). Contrary to the Hla toxin, the ACT toxin did not induce any cell detachment, or loss of cilia.

### 3.2. Treatment with the Hla or ACT Toxins Predisposes the Equine Respiratory Epithelium to EHV-1 Infection

#### 3.2.1. Infection of EHV-1 in the Respiratory Epithelium following α-Hemolysin Toxin Treatment

Figure 2A shows the effect of Hla pretreatment on EHV-1 infection at 24 h post inoculation (hpi).

Number of plaques: In nontreated control explants (i.e., submerged in serum-free medium), the number of viral plaques observed was 20 ± 11 per 4 mm^2^ epithelium. In Hla-treated explants, this number increased almost 4-fold to 78 ± 25 viral plaques per 4 mm^2^ epithelium (*p*-value < 0.05). EGTA pretreatment resulted in a significant increase in the number of plaques, compared to no treatment (84 ± 22). 

Plaque diameter: As for the plaque diameter, a similar trend was observed. Treatment with the Hla toxin increased the plaque diameter from 81 ± 9 µm in untreated explants to 118 ± 24 µm. Exposure to EGTA prior to EHV-1 inoculation led to a plaque diameter of 136 ± 7 µm. 

Percentage of infection: The overall percentage of EHV-1 infection in the epithelium is a combination of the total number of plaques and the average plaque diameter. The percentage of infection increased from 1.3 ± 0.4% in normal epithelium to 7.8 ± 3.7% after Hla treatment. In positive control explants (i.e., EGTA treated), the percentage of infection amounted to 11.1 ± 4.0%.

#### 3.2.2. Infection of EHV-1 in the Respiratory Epithelium following Adenylate Cyclase Toxin Treatment

Similarly to Hla treatment, exposure to the ACT toxin resulted in an upturn in EHV-1 infection in the epithelium at 24 hpi (Figure 2B).

Number of plaques: ACT toxin treatment led to a noticeable elevation in the number of plaques, as compared to the untreated control (34 ± 5.9 plaques in pretreated epithelium versus 11 ± 2.9 in the nontreated control; *p*-value < 0.05). 

The number of plaques in the positive control explant, treated with 8 mM EGTA, was markedly elevated as compared to the untreated explant (74 ± 9.9 plaques post EGTA treatment).

Plaque diameter: As for the plaque diameter, we observed a more than 2-fold incline after toxin treatment (148 ± 19.3 µm compared to 64 ± 8.0 µm; *p*-value < 0.01). In the positive control, a considerable incline in plaque diameter was noted, compared to the nontreated explant (190 ± 20.0 µm after EGTA exposure).

Percentage of infection: Clearly, the overall percentage of infection in the epithelium followed a similar trend as that of the number of plaques and plaque diameter. An elevation in the percentage of infection was noted after pretreatment with the ACT toxin. The percentage of infection after toxin treatment amounted to 5.1 ± 0.7%, as compared to 0.7 ± 0.3% in the untreated control (*p*-value = 0.06). In the positive control explant, we observed 14.2 ± 3.2% of infection in the epithelium. 

## 4. Discussion

The respiratory tract constitutes a primary portal of entry for many pathogens, including alphaherpesviruses. Consequently, it is not surprising that several innate barriers exist against these invading pathogens. Previously, our research group showed that the respiratory mucus layer is a crucial initial barrier to be overcome during the early pathogenesis of pseudorabies virus infection [31]. Whether mucus similarly acts as a first impediment for the closely related EHV-1 is not known. The key hurdle to be conquered by EHV-1 was shown to be the cell junctions. Indeed, Van Cleemput et al. recently demonstrated that EHV-1 targets a basolateral receptor, shielded from the apical environment by cell junctions [25]. However, the integrity of these junctions can be compromised by several environmental factors. When this happens, the basolateral receptor becomes freely accessible to the virus. Recently, it was demonstrated that respirable hazards, such as the mycotoxin deoxynivalenol, interfere with the structural integrity of cell junctions and thus predispose to EHV-1 infection [3]. Further, pollen proteases selectively and irreversibly destroy these epithelial junctions [2]. Finally, even the mucolytic drug lysomucil, often used in case of recurrent airway obstruction and severe pneumonia in horses, proves to have a similar disruptive effect on the epithelial barrier function [25]. Until now, exotoxins originating from bacterial pathogens were not investigated in this context. The present study aimed to investigate whether exotoxins originating from two respiratory bacteria in horses, *Staphylococcus aureus* and *Bordetella bronchiseptica,* could have a similar detrimental effect on epithelial integrity and EHV-1 infection. The question whether bacterial infections can predispose to viral infections may initially strike one as odd. After all, it is generally accepted that viral infections precede and promote bacterial colonization [32,33]. While a considerable amount of literature exists on the predisposing influence of viral infections on the occurrence of bacterial superinfections, substantially less literature is available on the reverse phenomenon [34]. Nevertheless, in the case of facultative pathogenic bacteria (such as *S. aureus*), or in the case of primary bacterial pathogens (*B. bronchiseptica*), it is relevant to question their potential to increase susceptibility to viral infection. The exotoxins that were investigated in this study include the α-hemolysin toxin from *S. aureus* and the adenylate cyclase toxin from *B. bronchiseptica.* Interestingly, previous studies pointed out that they disrupt cell junctions between epithelial cells of continuous cell lines (including Caco-2 and VA10 cells) [12,15]. Our hypothesis stated that these toxins have a similar effect on epithelial junctions in equine respiratory epithelium and thereby facilitate subsequent EHV-1 infection in the epithelium. To investigate our hypothesis, we used an in-house-developed ex vivo model: the equine respiratory mucosal explant model. Morphological analysis of the respiratory mucosa, based on HE staining of tissue sections, revealed that the integrity of the cell junctions remained unaltered upon treatment with both bacterial toxins. This is in striking contrast to previous studies performed on continuous cell lines, such as Caco-2 and VA10 cells, where treatment with the Hla and ACT toxins respectively caused a substantial impairment of the junctional proteins [12,15]. A possible explanation may rely on the fact that cell junctions in a monolayer of immortalized cells would adopt a more primitive form, as compared to the ex vivo explant situation, where epithelial cells and cell junctions are contained in a three-dimensional tissue structure. Both toxins induced a remodeling of the epithelium, characterized by a decrease in epithelial thickness. This finding is in agreement with results of an in vitro study that showed that the ACT toxin induces a reorganization of the actin skeleton in rat alveolar epithelial cells, resulting in an altered cell shape [35]. The cells lost their elongated appearance and instead acquired a round and compact shape. Remodeling of the actin skeleton in airway epithelial cells has also been described for the Hla toxin [14].

Analysis of the percentage of EHV-1 infection in the epithelium showed that the epithelium was predisposed to EHV-1 infection upon treatment with both bacterial toxins. An incline in the number of plaques as well as the plaque diameter was observed. An explanation for this observation could be that the compaction of epithelial cells might lead to a predisposition to EHV-1 infection (Figure 3). Certain junctional proteins, which are part of the cytoskeleton and are affected by the bacterial toxins, also modulate the localization of the tight junctions [36]. One of the main functions of these tight junctions is to prevent the migration of transmembrane proteins from the basolateral side of the cell membrane to the apical side [37]. Their altered localization might potentially result in the availability of the main EHV-1 receptor near the apical side of the respiratory epithelium. Future studies should investigate the latter hypothesis by performing direct virus-binding studies. Furthermore, in vivo experiments will need to be performed in the future to validate these results obtained with the respiratory mucosal explant system.

## 5. Conclusions

In conclusion, we are the first to demonstrate that two bacterial toxins, the α-hemolysin from *Staphylococcus aureus* and adenylate cyclase toxin from *Bordetella bronchiseptica*, sensitize the horse’s respiratory mucosa to EHV-1 infection. We propose that in the event a horse suffers from an infection with *S. aureus* or *B. bronchiseptica*, these bacteria may pave the way for a primary EHV-1 infection to occur. Our findings are directly relevant for the veterinary practitioner. So far, no successful curative therapies are available against EHV-1, and the current commercial vaccines have not been overly effective in the prevention of clinical disease. We therefore argue that the prevention of EHV-1 infections should always be combined with the control of environmental factors that may promote the onset of EHV-1 replication. Acknowledging the importance of predisposing factors and identifying them is a prerequisite in the prevention of EHV-1. Based on our findings, it should be recognized that bacterial infections with *S. aureus* or *B. bronchiseptica* could possibly precede an EHV-1 infection. These bacteria should thus at least be incorporated early on in the diagnostic approach of clinical respiratory disease. Upon diagnosing a predisposing infection with these bacteria, it may be beneficial to administer antimicrobial therapy early on in the course of the disease.

## Figures and Tables

**Figure 1 viruses-14-00149-f001:**
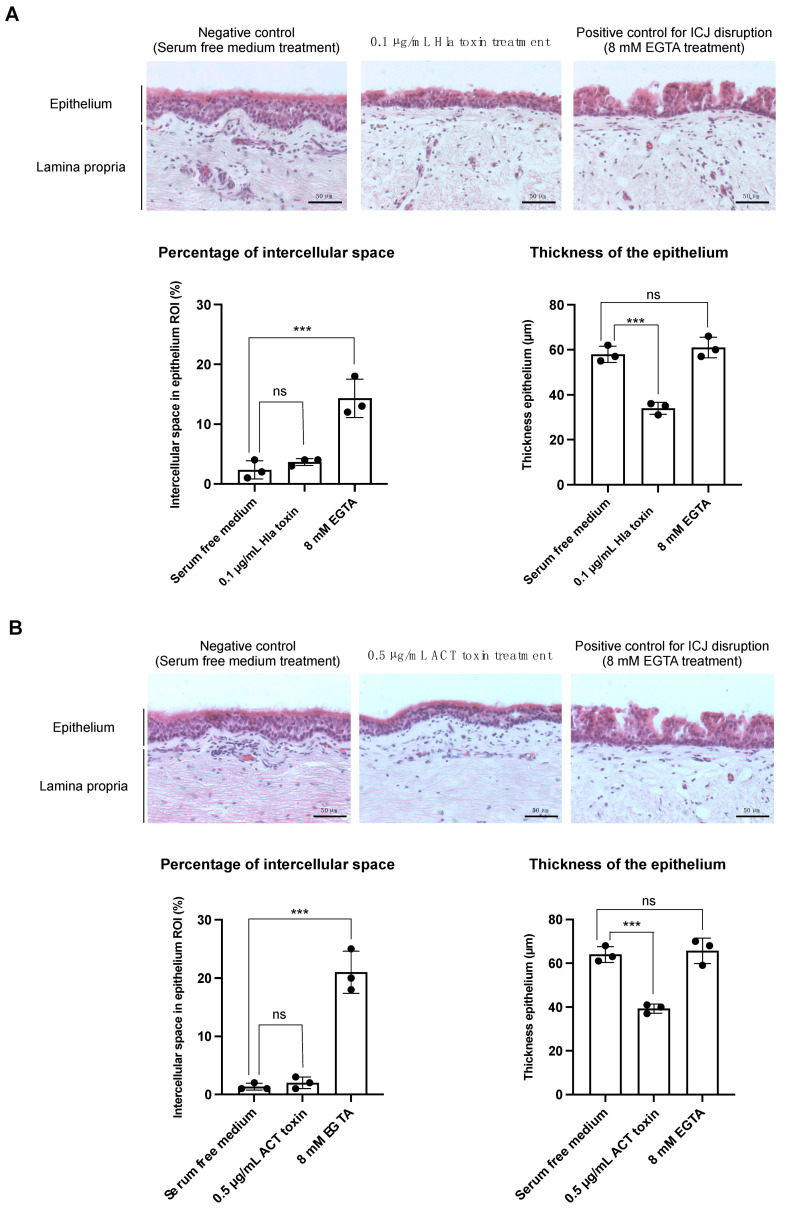
Treatment with the Hla and ACT toxins causes morphological changes in equine respiratory mucosa. (**A**) Effect of the Hla toxin on the morphology of equine respiratory mucosa. Hematoxylin-eosin (HE) stainings of paraffin-embedded sections of respiratory mucosal explants treated with Hla toxin or serum-free medium (**upper** panel). Central tendencies for intercellular space and epithelial thickness are indicated by the left and right bar plots, respectively (**lower** panel). (**B**) Effect of the ACT toxin on the mucosal morphology, as assessed by HE stainings (**upper** panel). The left and right bar plots show the percentage of intercellular space and the thickness of the epithelium, respectively (**lower** panel). Black dots in the bar plots represent independent replicates. Differences between means of the treatments were considered significant if the *p*-values were <0.05 (ns = not significant, *** = *p*-value < 0.001).

**Figure 2 viruses-14-00149-f002:**
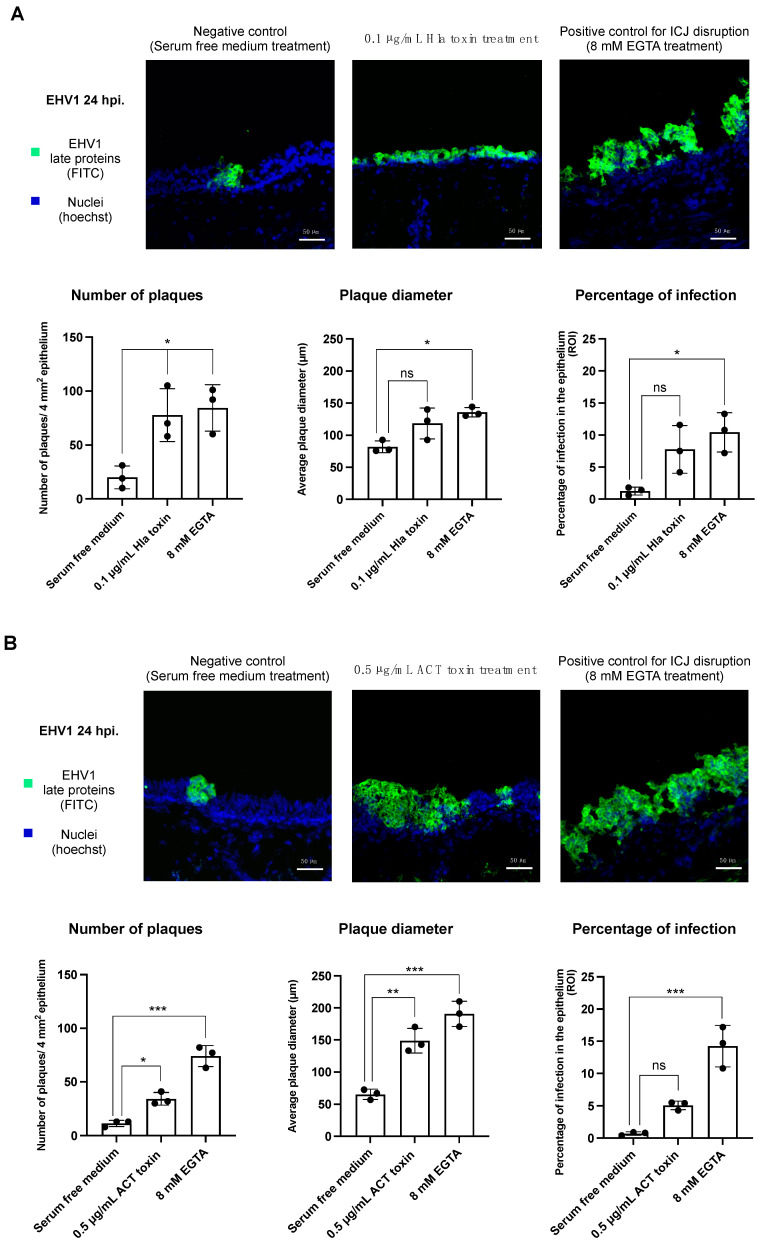
The effect of Hla and ACT toxin treatment on subsequent EHV-1 replication in the respiratory epithelium at 24 hpi. EHV-1 infection in the epithelium was evaluated by performing an immunofluorescent staining against EHV-1 late proteins on 50 consecutive cryosections/explants. The infection was evaluated by determining (1) the number of plaques/4 mm^2^ epithelium, (2) the average plaque diameter, and (3) the percentage of infection in the epithelium. (**A**) Effect of the Hla toxin on EHV-1 replication. Immunofluorescent stainings against EHV-1 late proteins in respiratory mucosal explants treated with Hla toxin or serum-free medium (**upper** panel). The left bar plot indicates central tendencies for the number of plaques. The plaque diameter is indicated by the middle bar plot. The bar plot on the right shows the percentage of infection in the epithelium (**lower** panel). (**B**) Effect of the ACT toxin on EHV-1 infection. The upper panel shows immunofluorescent stainings against EHV-1 late proteins in respiratory mucosal explants treated with the ACT toxin or with serum-free medium. The bar plots in the lower panel indicate central tendencies for the number of plaques (**left**), the plaque diameter (**middle**), and the percentage of infection in the epithelium (**right**). Black dots in the bar plots represent independent replicates. Differences between means of the treatments were considered significant if the *p*-values were <0.05 (ns = not significant, * = *p*-value < 0.05, ** = *p*-value < 0.01, *** = *p*-value < 0.001).

**Figure 3 viruses-14-00149-f003:**
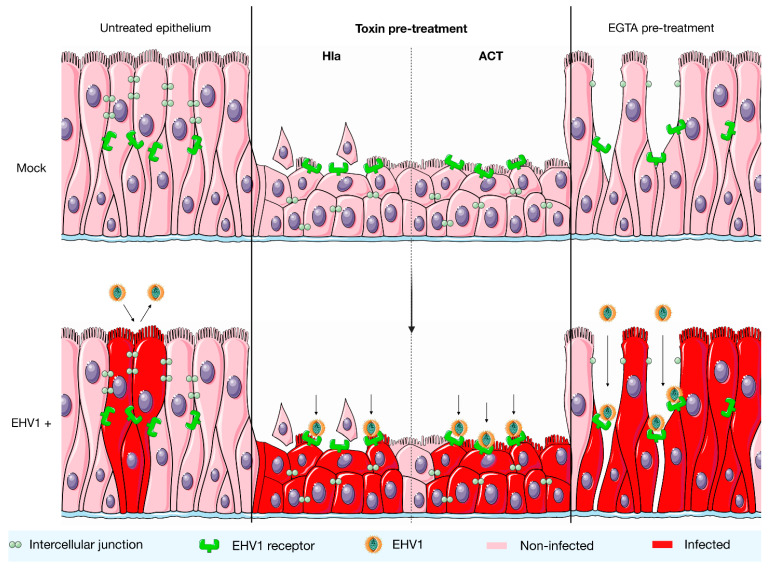
A hypothetical model on how exposure to Hla and ACT toxins might drive subsequent EHV-1 infection in the equine respiratory epithelium. Left: in normal healthy epithelium, EHV-1 replication is restricted due to the fact it targets a receptor located at the basolateral surface of the epithelial cells. This receptor is shielded from the apical environment by cellular junctions. Middle: after treatment with the Hla or ACT toxin, the epithelium undergoes morphological changes, characterized by a decrease in epithelial thickness. It may be that due to a remodeling of the actin cytoskeleton, the EHV-1 receptor becomes available near the apical side of the epithelium. This may explain the more efficient EHV-1 replication. Right: treatment with EGTA was included in this study as a positive control and leads to the disruption of cell junctions. Destruction of epithelial junctions drives EHV-1 infection by providing the virus free access to its basolateral receptor.

## Data Availability

Raw data are available from the corresponding author upon reasonable request.

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
