# Peer review of "Bacterial Toxins from Staphylococcus aureus and Bordetella bronchiseptica Predispose the Horse’s Respiratory Tract to Equine Herpesvirus Type 1 Infection"

_viruses, 2022, doi:10.3390/v14010149_

Round 1
Reviewer 1 Report
After reviewing the article titled "Bacterial toxins from Staphylococcus aureus and Bordetella bronchiseptica predispose the horse’s respiratory tract to equine herpesvirus type 1 infection", i would like to address comments and suggestion as below:
1- In the article, the equine herpesvirus type 1 is abbreviated (EHV1) or EHV-1; the authors are requested to use a same abbreviation in the paper. I suggest them to use the EHV-1 one.
2- The keywords are so long! the number of keywords needs to be reduced
3- In the last paragraph of the introduction section, line 76-77 and lines 83-84, the authors written about horses infection by S. aureus or B. bronchiseptica. Unfortunately, there is no information upon the prevalence of these bacteria among horses in the worlg and in Belgium. Two to three sentences can be added in the introduction to complete with related information.
4- Materials and methods:
2.1 The equine respiratory explant model
a) The autors are requested to provide right information concerning details on housing, husbandry and pain management of horses. Ethical consideration must be highlighted.
b) What is the time spent between the sampling and the transport of explants to the laboratory
c) Did the authors first evaluated the S. aureus and B. bronchiseptica resistance to the used antibiotics?
d) For how many time the tracheas were rinsed ? what was the quantity of the transport medium?
2.3. assessment of mucosal...viability
e) Line 144, the authors wrote "ImageJ". They nedd to separate Image and J.
2.4. EHV1 inclusion
f) Line 166: Why the authors decided to use the 6th passage of the viral culture?
2.5. Evaluation...microscopy
g) By choosing the dilution 1:20, did the authors test the others dilution and finally find that this one was the best one for this study? Could you explain how?
5- Results:
a) Line 274: 24 hpi. Could the authors define what is "hpi"?
b) Line 390-391: Figure 3. I was unable to find the reference to this figure in the article. Could the authors correct it?
6- In general:
There is no indication about: ethical, conflicts of interest, authors contribution, funding, etc...
Reviewer 2 Report
Comments to the Authors:
The authors aimed to evaluate the role of bacterial toxins from Staphylococcus aureus and Bordetella bronchiseptica in the alteration of the susceptibility of the epithelium of the upper respiratory tract to equine herpesvirus type 1. The given article attempts to validate the hypothesis that some bacterial infections may predispose to a viral infection. In doing so, the authors emphasize the novelty of their approach when reporting that bacterial exotoxins ‒ α-hemolysin and adenylate cyclase toxin increase the horse's sensitivity to equine herpesvirus type 1 infection. This study is partly based on the results obtained by the scientific group in the past. Unfortunately, the assessments are performed using exclusively light microscopy assays. Therefore, the correctness of some measurements raises doubt.
While this article does add to the growing data that bacterial toxins may contribute to the changes in the epithelium of the upper respiratory tract observed under exposure to equine herpesvirus type 1 infection, it offers much less convincing data to support this notion.
The authors should provide a point-by-point response to the reviewer's suggestions and improve the manuscript. A major revision is suggested.
Abstract
Is understandable to the reader. However, the disruption of cell junctions further accentuated in keywords as “intercellular junctions” is not pointed out.
Keywords
Too many keywords are used, half of them appear as arranged in phrases, others are too general (respiratory disease complexes).
The authors are recommended to use the term “cell junctions” instead of intercellular junctions throughout the text, which is a correct term when exploring cell biology issues. Furthermore, the authors are invited to use a common and uniform term instead of careless use of different terms (wrongly used), please see “intercellular connections”(L 206).
Introduction
The authors too extensively describe the results of their recent study (L49-53, L73-76) and use general phrases (L38-42, L59, L66-70), which are not of much interest. They are recommended to restructure the Introduction in a way, which facilitates reading, stratifying it into 4-5 paragraphs with a clear link to the title of the given article. The sentence “This impairment of the epithelial barrier system was shown to play an advantageous role in the early pathogenesis of the alphaherpesvirus equine herpesvirus type 1 (EHV1)” requires an additional word “infection”? Similarly, when discussing the pathogenesis of viral infections (L331).
L56 The phrase “EHV1 targets a basolateral receptor” should be corrected since a virus targets its receptor localized at the basolateral surface of the epithelial cell.
L63-64 The authors should rephrase the sentence “in the endothelial cells lining the blood vessels of the pregnant uterus”. More attention should be paid to the analysis of respiratory tract involvement when introducing this part to the readers.
A formulated aim does not appear at the end of the introduction part, it is replaced by a scientific hypothesis.
Materials and Methods
Section 2.2. L 131 I suppose the authors intended to assess the sections (not stainings!) stained with hematoxylin and eosin. The authors should consider rephrasing. Similarly, “stained with an HE staining” (L139). The aforementioned sentence is suboptimal.
Section 2.3.
The thickness of the histological section may affect the results of the assessment. The authors are asked to explain why such thick sections (sections of 8 μm) were cut?
When the measurements of dilated intercellular spaces appear as a routine procedure conducting ultrastructural analysis, the accuracy of these measurements dramatically decreases when operating at the tissue level. Furthermore, an application of a complex and prolonged tissue (explant) processing procedure brings additional artifacts due to a fixative-induced cell/tissue shrinkage. The magnification used to perform an analysis of the sections is not included in the methodology description (similarly, when describing the TUNEL reaction assessment), but based on the figures attached with scale bars, it is not high. Few papers have reported on the measurements of dilated intercellular spaces observed in the esophagus epithelium as far. The methodology of assessment is not generally accepted. All aforementioned raises doubts on the correctness of measurements performed. Therefore, at this particular point, the authors need to find a robust explanation and develop a convincing argument.
Section 2.5.
Routinely, the cryostat sections are fixed or double fixed with cold acetone. Please explain, why the sections mounted on slides were “postfixed” using 4% paraformaldehyde. Moreover, cryostat tissue sections are quite vulnerable concerning Triton-X100. Please specify the necessity of this application.
The authors are asked to check the correctness of the description when specifying the immunofluorescence staining method (L179-187).
Please name the manufacturer of the primary anti-EHV1 antibody, specify correctly what does it recognize?
The phrase “The percentage of infection in the epithelium (i.e. ROI)” (L190) reflects a poor understanding of the basic principles of antigen detection by the use of immunohistochemistry.
Results
TUNEL assay is used to assess the DNA fragmentation characteristic of apoptosis. Consider rephrasing “treatment… did not result in toxicity at the level ? of the respiratory epithelium or the lamina propria, by means of TUNEL stainings” (L201-204).
Section 3.1. “The Hla and ACT toxins cause a reduction in epithelial thickness, but do not alter the integrity of intercellular connections in equine respiratory mucosa”. Since the authors have not been studying cell junctions, the title of the given section has no grounds and needs to be removed.
Sections 3.1.1 and 3.1.2.(and Figure 1 as well) The titles when specifying “Morphological effects…” need to be rephrased. Simultaneously, the authors should repeatedly review the recommendations, which appear under the “Keywords”.
The use of well-known terms and phrases with a particular application always occurs with a certain caution. The term “an infection or an incidence rate” is typically used to measure the frequency of occurrence of new cases of infection within a defined population during a specified time frame. It is a useful parameter when evaluating the rate at which a disease is spread among people or animals. Since the application of generally recognized terminology differs in the given study, the authors should explain it.
Figure 2. The authors should show a better understanding of the principles of the expression of antigen/antigens (not clearly described in Materials and methods) and its assessment when addressing the issues of immunofluorescence.
Figure 3. While this drawing does add to the readers’ understanding that bacterial toxins may severely affect the structure of the epithelial cells, it offers excessive speculative data (an actin cytoskeleton remodeling, the translocation of the EHV1 receptor, and changes in the molecular constituents of cell junctions). The aforementioned issues were not explored in the given study. The pseudostratified epithelium belongs to a group of simple epithelia. Does it undergo so a dramatic change (stratified double-layered epithelium) under the toxin exposure (pre-treatment as it is shown in the middle)? The changes in the cellular organization and architecture were not assessed by the authors. Finally, epithelial dysplasia appears only briefly mentioned (L371) when discussing the changes in morphology.
Discussion
Is well-written and is enough attractive for the reader.
Actin is a ubiquitous protein that always contributes to the changes in a cell shape when remodels. Please give reasons for applying the actin cytoskeleton description. Moreover, the alveoli lining differs from that observed in the upper respiratory tract [reference 37].
L382-383 sounds like speculation and should be removed. While the possible future step of the EHV1-related research (vaccination, treatment) in horses are highlighted, the limitations of the study are not written.
Citation
The citations N21 and N27 should be corrected.
Linguistic suggestions
Minor language editing should be performed.

Round 2
Reviewer 2 Report
Overall, the authors have followed the recommendations addressed in the 1st round of revision and have provided a point-by-point response.
The reviewer’s point of view when assessing the 14th response: routinely performed things do not necessarily always act correctly. It is a weak argument. By the use of immunohistochemistry (either bright-field optics or fluorescence), one detects the expression of tissue antigen. You have explored the expression of EHV-1 late proteins. Therefore, when stating clearly (when compared to the percentage of infection) the authors may expect the facilitation of the reader’s understanding.
When striving for an ideal manuscript attractive for the wider readership an author should peruse the text thoroughly. Therefore, some minor (linguistic) changes are recommended to the 1-2 sentences (the 1st paragraph) to avoid the repeated use of words (respiratory: respiratory symptoms, hazards, mucosa, pathogens).
Similarly and repeatedly, please rephrase the pregnant uterus (L73-74) as a condition referred to a human/animal, not the organ.
Final decision: the manuscript was improved and after some minor corrections to it, could be suitable for publication.
Overall, the authors have followed the recommendations addressed in the 1st round of revision and have provided a point-by-point response.
The reviewer’s point of view when assessing the 14th response: routinely performed things do not necessarily always act correctly. It is a weak argument. By the use of immunohistochemistry (either bright-field optics or fluorescence), one detects the expression of tissue antigen. You have explored the expression of EHV-1 late proteins. Therefore, when stating clearly (when compared to the percentage of infection) the authors may expect the facilitation of the reader’s understanding.
When striving for an ideal manuscript attractive for the wider readership an author should peruse the text thoroughly. Therefore, some minor (linguistic) changes are recommended to the 1-2 sentences (the 1st paragraph) to avoid the repeated use of words (respiratory: respiratory symptoms, hazards, mucosa, pathogens).
Similarly and repeatedly, please rephrase the pregnant uterus (L73-74) as a condition referred to a human/animal, not the organ.
Final decision: the manuscript was improved and after some minor corrections to it, could be suitable for publication.
